# Neural Radiance Field-Inspired Depth Map Refinement for Accurate Multi-View Stereo [note 1]

**DOI:** 10.3390/jimaging10030068

**Published:** 2024-03-08

**Authors:** Shintaro Ito, Kanta Miura, Koichi Ito, Takafumi Aoki

**Affiliations:** Graduate School of Information Sciences, Tohoku University, 6-6-05, Aramaki Aza Aoba, Sendai 9808579, Japan; kanta@aoki.ecei.tohoku.ac.jp (K.M.); aoki@ecei.tohoku.ac.jp (T.A.)

**Keywords:** multi-view stereo, neural radiance fields, depth map estimation, 3D reconstruction

## Abstract

In this paper, we propose a method to refine the depth maps obtained by Multi-View Stereo (MVS) through iterative optimization of the Neural Radiance Field (NeRF). MVS accurately estimates the depths on object surfaces, and NeRF accurately estimates the depths at object boundaries. The key ideas of the proposed method are to combine MVS and NeRF to utilize the advantages of both in depth map estimation and to use NeRF for depth map refinement. We also introduce a Huber loss into the NeRF optimization to improve the accuracy of the depth map refinement, where the Huber loss reduces the estimation error in the radiance fields by placing constraints on errors larger than a threshold. Through a set of experiments using the Redwood-3dscan dataset and the DTU dataset, which are public datasets consisting of multi-view images, we demonstrate the effectiveness of the proposed method compared to conventional methods: COLMAP, NeRF, and DS-NeRF.

## 1. Introduction

Multi-View Stereo (MVS) is a technique for acquiring 3D data from target objects or scenes from multiple images captured by a camera [1,2,3]. Since MVS requires only camera images, it is not restricted to the capturing environment, reduces the effort required for capturing images, and can more easily acquire 3D data compared to active scanners.

MVS estimates depth maps from images taken from different viewpoints and integrates them to reconstruct 3D data [2,3,4,5,6,7]. A depth map is an image in which the pixel values represent the distance, i.e., the depth, from the camera to the object. To estimate the depth map for each viewpoint, MVS performs image matching between multi-view images. One of the typical methods is plain sweeping [4,8]. In plain sweeping, the most optimal depth is searched for in each pixel of the input image based on the similarity of textures in the local region of the image while varying the depth from the camera to the object, where Normalized Cross-Correlation (NCC) or Zero-mean Normalized Cross-Correlation (ZNCC) between local regions is generally used as the similarity [3,8]. Although the optimal depth can be estimated by taking into account the geometric consistency among multi-view images, the number of image-matching operations becomes large since a full depth search is required for each pixel [7]. To reduce the number of image matching operations in MVS, efficient methods using PatchMatch [9,10] have been proposed [3,7,11,12]. Among them, COLMAP [3,13] has been proposed as a pipeline for 3D reconstruction using PatchMatch and is used as a de facto standard method for MVS. PatchMatch-based methods assign depth and normal as parameters to each pixel. For example, the initial value of the depth is a random number within the acceptable range of depth estimation obtained from the epipolar constraints between cameras, and the initial value of the normal is a random number within ±π/3 for the angles of the *X* and *Y* axes [7]. Then, the parameters are optimized by matching corresponding pixels in different viewpoints according to these parameters. The depth map can be estimated with fewer matching operations than a full search by using random numbers as the initial values of the parameters. Since the parameters are optimized using image matching, the accuracy of depth estimation is degraded in poor-texture regions and at object boundaries, and occlusion prevents depth estimation. Recently, depth map estimation methods using deep learning have been proposed [14,15,16,17]. In this paper, “texture” refers to the spatial distribution of colors and their intensities in an image. “Rich texture” indicates that there is a large difference in intensity values between pixels and that the texture has a complex pattern, while a “poor texture” indicates that there is almost no difference in intensity values between pixels and that the texture is uniform. “Object boundary” indicates the boundary between the foreground and background in the image, where the pixels have significantly different depths. A typical method, namely MVSNet [15], projects feature maps extracted by a Convolutional Neural Network (CNN) [18] to another viewpoint based on plain sweeping, and estimates the depth of each pixel based on the similarity of the features. Depth map estimation with training is more accurate than that without training since CNN-based methods can use features considering the shape and positions of the neighboring regions of the pixel of interest as well as textures. On the other hand, even with deep learning, depth estimation is difficult in poor-texture regions and object boundaries. Thus, the depth map estimation in MVS can accurately estimate the depth on object surfaces with rich texture, while the estimation accuracy is degraded in poor-texture regions and at object boundaries.

Neural Radiance Fields (NeRFs) [6] have been proposed as another method for depth map estimation from multi-view images. NeRF represents a 3D space as a radiance field, which is parametrized with a Multi Layer Perceptron (MLP). The MLP is trained so as to estimate a volume density and view-dependent emitted radiance given the spatial location and view direction of the camera from multi-view images. The use of the trained MLP makes it possible to synthesize images from novel viewpoints based on the radiance field on the ray connecting the camera and the object. NeRF can not only generate novel view images from the radiance field, but can also generate depth maps. Depth can be synthesized pixel by pixel using the radiance field, even for poor-texture regions and object boundaries. On the other hand, it is not always possible to accurately estimate the depth on the surface of an object using NeRF compared with MVS.

As described above, MVS estimates depths based on image matching and thus can accurately estimate depths on object surfaces with rich texture, while the accuracy of depth estimation is degraded in poor-texture regions and at object boundaries. On the other hand, NeRF estimates the radiance field of a scene from multi-view images and estimates depth for each pixel from the radiance field, and thus can estimate depth for poor-texture regions and object boundaries, while the accuracy is not always high for object surfaces. In this paper, we propose a method to refine the depth maps obtained by MVS through the iterative optimization of NeRF. The standard NeRF trains an MLP to generate novel view images, while the proposed method refines the depth map by iteratively optimizing an MLP, so that the MLP can render the input image and the depth map obtained by MVS. Therefore, the proposed method only performs iterative optimization of the radiance field and does not require any training. Through a set of experiments using the Redwood-3dscan dataset [19] and the DTU dataset [20], which are public datasets consisting of multi-view images, we demonstrate the effectiveness of the proposed method compared to conventional methods. In the experiments, we employ an evaluation metric that is invariant to the depth scale [21], in addition to the widely used evaluation metrics for depth map estimation.

## 2. Related Work

This section summarizes the depth map estimation methods using MVS and NeRF that are related to this study.

### 2.1. MVS-Based Approaches

Here, we give an overview of COLMAP [3,13] using PatchMatch and MVSNet [15] using deep learning as MVS-based depth map estimation methods.

#### 2.1.1. COLMAP

COLMAP is a pipeline for 3D reconstruction from multi-view images that consists of Structure from Motion (SfM) [13] and MVS [3]. SfM is a method for 3D reconstruction and camera parameter estimation by sequentially adding images using the principle of triangulation used in stereo vision [1]. Correspondence point pairs are obtained based on the similarity between feature points, 3D points are reconstructed using the correspondence point pairs and camera parameters based on the principle of triangulation, and camera parameters are optimized by minimizing reprojection errors. SfM estimates camera parameters and reconstructs sparse 3D point clouds from multi-view images. SfM in COLMAP is a de facto standard method among many MVS methods for estimating the camera parameters of multi-view images. MVS estimates the depth map of each viewpoint using the results of SfM and reconstructs dense 3D point clouds. MVS in COLMAP, similar to PatchMatch, assigns depth and normal to each pixel as parameters initialized with random numbers, and then iteratively performs image matching and parameter propagation among multi-view images to optimize the depth and normal. To improve the accuracy of depth map estimation using PatchMatch, MVS in COLMAP utilizes the following ideas: (i) propagates parameters taking into account the geometry by selecting the pixel of interest and the corresponding view for each pixel based on the camera rotation, occlusion obstructing the view, and image resolution, (ii) employs NCC with bilateral weights for image matching in local regions, (iii) improves the accuracy of depth estimation by maximizing the photometric consistency and minimizing the geometric consistency based on reprojection errors between viewpoints, and (iv) removes outliers according to the confidence value calculated by the photometric consistency and the geometric consistency. COLMAP also has problems with low depth estimation accuracy in poor-texture regions and at object boundaries.

#### 2.1.2. Deep Learning

Recently, a number of depth map estimation methods using deep learning have been developed [14,15,16,17,22]. Here, we describe one of the typical methods, MVSNet [15]. MVSNet estimates a depth map for each viewpoint through three steps: feature extraction from multi-view images, the creation of a cost volume, and depth map estimation. Let the image for which the depth map is to be obtained be the reference image, and the images in the neighborhood of the reference image be the neighboring images. Feature maps are extracted from both the reference image and the neighboring images using a 2D CNN. A virtual plane is assumed in the depth direction of the camera for the reference image, the feature maps of the neighboring images are projected onto the virtual plane by homography transformation, a feature volume at each viewpoint is created, and a scene cost volume is created by aggregating the feature volumes of the reference image and the neighboring images. The cost volume is used to determine the existence probability of object surfaces in the depth direction, and the depth of each pixel is estimated from its expected value.

As described above, MVS estimates the depth map using image matching based on the texture in the images and the features extracted by CNN. Because of the use of image matching, the depth map can be estimated with high accuracy in rich-texture regions, while the estimation accuracy degrades in poor-texture regions and at object boundaries. In addition, MVS is difficult to estimate the depth in regions containing occlusions even though the deformation between images is normalized using a homography transformation to improve the accuracy of image matching.

### 2.2. NeRF-Based Approaches

We describe a novel view synthesis method, i.e., NeRF [6] and depth map estimation using NeRF. We also describe Depth-Supervised NeRF (DS-NeRF) [23], which utilizes sparse 3D point clouds reconstructed by SfM, as a depth map estimation method using NeRF.

#### 2.2.1. NeRF

NeRF estimates the radiance field of a 3D scene from multi-view images and camera parameters using an MLP, and synthesizes a novel view by volume rendering [24] the radiance field [6]. The MLP takes the coordinates x=(x,y,z) of a 3D point in its direction (θ,ϕ) as the input and the RGB value c=(r,g,b) of the 3D point and the density σ representing the opacity of the 3D point as the output. The ray $r_{i}$ from the camera center o in the camera image *I* through the pixel *i* in the camera image *I* and the 3D point $x_{i}$ corresponding to the pixel i∈I is defined by
(1)$$r_{i}\left(t\right)=o+td_{i},$$
where *t* is the position on the ray and $d_{i}$ is its direction $\left(\theta_{i},\phi_{i}\right)$ which observes the 3D point x. From the RGB value $c(r_{i}\left(t\right),d_{i})$ of a 3D point on the ray and the density $\sigma (r_{i}\left(t\right))$ of 3D points, the pixel value $C_{i}$ at pixel *i* is calculated by
(2)$$C_{i}=\int_{t_{\mathrm{near}}}^{t_{\mathrm{far}}}T_{i}\left(t\right)\sigma \left(r_{i}\left(t\right)\right)c\left(r_{i}\left(t\right),d_{i}\right)\;dt,$$
where $t_{near}$ and $t_{far}$ indicate the range of volume rendering and $T_{i}\left(t\right)$ is an accumulated transmittance function, which describes the phenomenon that the brightness of rays is attenuated by objects, and is defined by
(3)$$T_{i}\left(t\right)=\exp\left(-\int_{t_{\mathrm{near}}}^{t}\sigma (r_{i}\left(s\right))\;ds\right).$$

In practice, since *N* 3D points on the sampled rays $\hat{r}$ are used, Equation (Equation 2) can be rewritten as
(4)$$\hat{C}\left(\hat{r}\right)=\sum_{j=1}^{N}T_{j}(1-\exp\left(-\sigma_{j}\delta_{j})\right)c_{j},$$
where $\delta_{j}=t_{j+1}-t_{j}$ denotes the distance between adjacent 3D points located on the ray and $T_{j}$ is given by
(5)$$T_{j}=\exp\left(-\sum_{k=1}^{j-1}\sigma_{k}\delta_{k}\right).$$

The MLP is trained with the loss function L between the pixel values $\hat{C}\left(\hat{r}\right)$ of the image synthesized by volume rendering and $C^{gt}\left(\hat{r}\right)$ of the camera image, which is defined by
(6)$$L=\sum_{\hat{r}\in R}{\left|\right|\hat{C}\left(\hat{r}\right)-C^{gt}\left(\hat{r}\right)\left|\right|}^{2},$$
where R is a set of rays passing through each pixel. In NeRF, the depth $D(\hat{r})$ is calculated by using the density σ of sampled 3D points on the ray $\hat{r}$ and $T_{i}$ obtained from the density [25,26,27,28] as follows:(7)$$D\left(\hat{r}\right)=\sum_{j=1}^{N}T_{j}\left\{1-\exp\left(-\sigma_{j}\delta_{j}\right)\right\}t_{j}.$$

A depth map for each viewpoint can be obtained by calculating the depth for all the pixels. NeRF does not use image matching for local regions, and therefore can estimate depth maps with high accuracy in poor-texture regions and at object boundaries.

#### 2.2.2. DS-NeRF

Here, we describe Depth-Supervised NeRF (DS-NeRF) [23], which combines NeRF and SfM in COLMAP as a method to improve the performance of NeRF. As mentioned above, NeRF trains an MLP using the color reconstruction loss between the synthesized image and the camera image. In addition to the color reconstruction loss, DS-NeRF uses the depth loss between the depth obtained by volume rendering and the depth obtained from the sparse 3D point cloud in SfM. DS-NeRF can train an MLP more efficiently than NeRF and can synthesize novel views from a small number of images. The depth loss $L_{Depth}$ used in DS-NeRF is calculated based on KL divergence as follows:(8)$$L_{Depth}\approx E_{x_{i}\in X_{j}}\sum_{k}\log h_{k}\exp\left\{-\frac{{\left(t_{k}-D_{ij}\right)}^{2}}{2{\hat{\sigma }}_{i}^{2}}\right\}\Delta t_{k},$$
where $X_{j}$ indicates a set of feature points visible from camera *j*, $x_{i}$ indicates the *i*-th feature point, $h_{k}$ indicates the existence probability of the object surface at the *k*-th sampling point on the ray, ${\hat{\sigma }}_{i}$ indicates the reprojection error at the *i*-th feature point $x_{i}$, and $D_{ij}$ indicates the distance from camera *j* to the feature point *i*. The larger the reprojection error of the feature points, the weaker the loss constraint is to take into account the estimation error of the 3D points by SfM. Although depth maps can be estimated from a small number of images, sparse depth maps have to be used for training in DS-NeRF. Therefore, it is not always possible to synthesize a highly accurate depth map by volume rendering using the radiance field.

Recently, RC-MVSNet [17] has been proposed, which combines CasMVSNet [22] and NeRF to train CasMVSNet by unsupervised learning. Although unsupervised learning reduces the limitation on the amount of training data, the depth map cannot always be estimated with high accuracy since NeRF is estimated based on the depth map generated by CasMVSNet.

## 3. NeRF-Inspired Depth Map Refienment

As mentioned above, the depth map estimated by MVS does not obtain depth in poor-texture regions, occlusions, and at object boundaries. We propose a depth map estimation method multi-view images with NeRF-inspired depth map refinement. The proposed method differs from general NeRF in that it iteratively optimizes the MLP to synthesize the input image and the depth map estimated by MVS, rather than training the MLP to synthesize novel view images. NeRF trains the radiance field of the scene using the input multi-view images and uses it to synthesize novel view images. On the other hand, the proposed method refines the depth map by optimizing the radiance field of the scene so that the input multi-view images and the dense depth map can be synthesized. The proposed method corresponds to overfitting the training data from the viewpoint of NeRF. Since NeRF aims to synthesize novel view images, while the proposed method aims to refine the input depth maps, the proposed method can achieve its objective even by overfitting the training data in NeRF. In the following, we refer to “optimize” as overfitting the MLP to the training data to estimate a depth map from the same viewpoint as the training data. We also refer to “train” as synthesizing a novel view by training the MLP with the training data, i.e., normal NeRF. We describe an overview of the proposed method, the network architecture of the MLP used in the proposed method, and the objective functions for optimization in the following.

### 3.1. Overview

The proposed method consists of camera parameter estimation by SfM, depth map estimation by MVS, and depth map refinement by NeRF optimization, taking multi-view images as the input. The framework of the proposed method is shown in Figure 1, which is inspired by DS-NeRF [23]. DS-NeRF uses sparse 3D point clouds obtained by SfM to train the MLP so as to synthesize novel views using NeRF. On the other hand, the proposed method refines the depth map obtained by MVS through the optimization of an MLP, which is different to DS-NeRF. The proposed method uses COLMAP to estimate the camera parameters [13] and depth maps [3] to compare the performance of the proposed method with that of DS-NeRF. Therefore, it should be noted that the COLMAP process can be replaced by other SfM and/or MVS methods in the proposed method. The proposed method iteratively optimizes the MLP representing the radiance field using the dense depth map estimated by MVS. We optimize the MLP so that the depth map is synthesized by volume rendering to be close to the depth map estimated by MVS, and so that the image from the same viewpoint as the input image is synthesized. As a result, it is possible to estimate the depth in poor-texture regions and at object boundaries that cannot be estimated by MVS. We obtain a depth map that is more accurate than MVS by volume rendering the depth map using the optimized MLP.

### 3.2. Network Architecture of an MLP

An MLP, which refines the depth maps obtained by MVS, consists of the network architecture as shown in Figure 2. This network architecture is designed based on DS-NeRF [23]. A 3D point x=(x,y,z) and its direction d=(ϕ,θ) are inputs, and RGB values c and the density σ of x are outputs. Three-dimensional points x and view direction d are applied during positional encoding [6] to create higher dimensional vectors γ(x) and γ(d), which are input to the MLP. We generate 256-dimensional feature vectors passing γ(x) through eight fully-connected layers with the ReLU activation function. The output of the fifth layer is concatenated with γ(x) using skip connection. Then, the 3D point density σ and 256-dimensional feature vectors are obtained by passing them through a fully-connected layer. The output feature vector is then concatenated with the feature vector γ(d), and the RGB values of the 3D points are output through a fully connected layer.

### 3.3. Objective Functions

We describe the objective functions that are required in the optimization of the MLP to refine the depth maps obtained by MVS. Note that we use the term “loss” in the following since the only differences between the loss function used in training the MLP and the objective function used in MLP optimization are the expressions “loss” and “error”. The proposed method employs the color reconstruction loss $L_{Color}$ [6] as the objective function for color reconstruction and the depth loss $L_{Depth}$ based on Huber loss [29] as the objective function for depth reconstruction.

#### 3.3.1. Color Reconstruction

The color reconstruction loss, $L_{Color}$, is the mean squared error loss between the pixel values estimated by volume rendering using Equation (Equation 4) and the pixel values of the same pixel in the input image and is defined by
(9)$$L_{Color}=\sum_{j\in J}\left|\right|C_{j}-C_{j}^{gt}{\left|\right|}^{2},$$
where *J* indicates a set of pixels in the input image, $C_{j}$ indicates pixel values synthesized by volume rendering at pixel *j*, and $C_{j}^{gt}$ indicates pixel values of the same pixel in the input image.

#### 3.3.2. Depth Reconstruction

We propose a new loss function based on Huber loss [29] for depth reconstruction that is robust against outliers. We consider that it is important to have robustness against outliers since the depth maps obtained by MVS in COLMAP contain many outliers. Huber loss is a loss function that combines L1 loss and L2 loss. Using the idea of Huber loss, the proposed method uses the mean squared error loss, i.e., L2 loss, when the error between the depth obtained by volume rendering and the depth obtained by MVS in COLMAP is smaller than a threshold ϵ, and the absolute error loss, i.e., L1 loss, when the error is larger than ϵ. The term $H(D_{k},D_{k}^{mvs})$ based on Huber loss used in the depth loss $L_{Depth}$ is defined by
(10)$$H\left(D_{k},D_{k}^{mvs}\right)=\left\{\begin{array}{cc} \frac{a^{2}}{2} & |a|\leq \epsilon \\ \epsilon \left(\left|a\right|-\frac{\epsilon }{2}\right) & \mathrm{otherwise} \end{array}\right.,$$
where $D_{k}$ indicates the depth at pixel *k* obtained by volume rendering, $D_{k}^{mvs}$ indicates the depth at pixel *k* in the depth map estimated by MVS in COLMAP, $a=D_{k}-D_{k}^{mvs}$, and $\epsilon =\frac{t_{far}-t_{near}}{N_{coarse}-1}$. As mentioned above, the depth is obtained by accumulating the densities of 3D points on the rays in volume rendering. The error between the depth obtained by volume rendering and the depth obtained by MVS in COLMAP should be smaller than the distance between adjacent 3D points. Therefore, we use the number of sampling points $N_{coarse}$ used for coarse sampling in hierarchical volume sampling as the threshold ϵ. Then, the depth loss $L_{Depth}$ used in the proposed method is defined by
(11)$$L_{Depth}=\frac{1}{K}\sum_{k\in K}H(D_{k},D_{k}^{mvs}),$$
where *K* indicates a set of pixels in the input image whose depth $D_{k}^{mvs}$ is obtained by MVS in COLMAP. Note that the depth loss $L_{Depth}$ is calculated only for pixels with depth obtained by MVS in COLMAP.

The iterative optimization of the MLP used in the proposed method employs an objective function that combines the color reconstruction loss and depth loss described above, which is given by
(12)$$L=L_{Color}+\lambda_{D}L_{Depth},$$
where $\lambda_{D}$ indicates a hyper parameter.

## 4. Experiments and Discussion

This section describes experiments to evaluate the accuracy of the proposed method using public datasets of multi-view images. We describe the dataset used in the experiments, the experimental conditions, evaluation metrics, ablation study of depth loss, accuracy comparison with conventional methods, and 3D reconstruction in the following.

### 4.1. Dataset

We describe two multi-view image datasets, i.e., the Redwood-3d scan dataset (https://redwood-data.org/3dscan/index.html (accessed on 7 February 2024)) [19] and the DTU dataset (https://roboimagedata.compute.dtu.dk/?page_id=36 (accessed on 7 February 2024)) [20], which are used in the experiments.

#### 4.1.1. Redwood-3d Scan Dataset (Redwood)

Redwood consists of 10,933 RGB-D video images taken in a variety of scenes and 441 3D mesh models. There are 44 different categories of scenes, such as chairs, tables, sculptures, and plants. The RGB-D video images were taken by non-experts in computer vision, and many of them contain low-quality frames and poor-texture regions. Therefore, it is difficult to reconstruct 3D shapes from the multi-view images in Redwood using MVS due to external factors such as motion blur, noise, poor-textured objects, and illumination changes. In our experiments, we use 12 scenes: “amp#05668”, “chair#04786”, “chair#05119”, “childseat#04134”, “garden#02161”, “mischardware#05645”, “radio#09655”, “sculpture#06287”, “table#02169”, “telephone#06133”, “travelingbag#01991”, and “trashcontainer#07226” as shown in Figure 3. We extract 11 frames from the RGB-D video image of each scene, and use the RGB image with 640 × 480 pixels of each frame as the input and the depth map as the ground truth for accuracy evaluation. The camera parameters for each viewpoint used in all the depth map estimation methods are estimated by SfM in COLMAP [13].

#### 4.1.2. DTU Dataset (DTU)

DTU consists of multi-view images of a variety of objects, a 3D point cloud measured by a laser scanner, and the camera parameters. The multi-view images consist of a set of images with 1600 × 1200 pixels, which are taken of each object from 49 or 64 viewpoints. Multi-view images in DTU are acquired under the controlled environment. Therefore, we can evaluate the potential performance of MVS methods themselves since there are few external factors using DTU. There are 124 types of objects, such as building models, animal figurines, plants, and vegetables. We use the “scan9”, “scan33”, and “scan118” as shown in Figure 4. Due to the processing time, we resize the images to 800 × 600 pixels and use them as input images. Since the images in DTU were taken under seven different lighting conditions, we use the multi-view image taken under one of the seven lighting conditions. The camera parameters for each view used in all the depth map estimation methods are estimated by SfM in COLMAP [13]. Since DTU does not have the ground truth for evaluating the accuracy of the depth map estimation, we use the depth maps created by Yao et al. [5,15].

### 4.2. Experimental Condition

In our experiments, we compare the accuracy of depth map estimation among COLMAP [3], NeRF [6], DS-NeRF [23], RC-MVSNet [17], and the proposed method to demonstrate the effectiveness of the proposed method. NeRF and DS-NeRF train an MLP that represents the radiance field using multi-view images so that novel view images can be synthesized. By inputting a novel view direction to the trained MLP, the image and depth map of that view can be synthesized. NeRF and DS-NeRF need to train an MLP using training data and evaluate it on test data. On the other hand, the proposed method optimizes an MLP that represents the radiance field so that the input images and depth maps can be synthesized. In this experiment, we estimate depth maps for the input known viewpoints. To evaluate the accuracy under the same conditions as the proposed method, NeRF and DS-NeRF trained an MLP using the input multi-view images and use the trained MLP to synthesize depth maps for the input multi-view images. Therefore, we trained NeRF and DS-NeRF a certain number of times as in the proposed method. Table 1 shows the hyper parameters used in the experiments. The number of training or optimization iterations was set to 15,000 for Redwood and 100,000 for DTU, since the number of images and the number of pixels are different for each dataset. The batch size, which represents the number of rays in each iteration, was set to 5120. DS-NeRF and the proposed method, which require the depth map loss to be calculated, have a parameter $\lambda_{D}$ that controls the ratio of depth rays used to calculate the depth loss within the batch size and the weights of the loss function. The ratio of depth rays used in DS-NeRF and the proposed method are set to 0.5 and 0.2, respectively, and $\lambda_{D}$ is set to 0.1 for both methods. NeRF, DS-NeRF, and the proposed method use hierarchical volume sampling [6] as a sampling method based on the density of points on a ray. Hierarchical volume sampling first produces the color and density of $N_{coarse}$ 3D points in a coarse network, and then produces the color and density of $N_{fine}$ 3D points belonging to high-density regions in a fine network. We set $N_{coarse}=64$ and $N_{fine}=N_{coarse}+128$ for all the methods in the experiments. In the experiments, Adam [30] is used as the optimizer. The learning rate begins at $5.0\times 10^{-4}$ and decays exponentially to $5.0\times 10^{-5}$ during the optimization process. For RC-MVSNet, we use the trained model and evaluation code available in the official GitHub repository (https://github.com/Boese0601/RC-MVSNet (accessed on 26 Feburary 2024)). The threshold for the reprojection error used in depth map filtering is set to 0.5 pixels.

### 4.3. Evaluation Metrics

We evaluate the accuracy of depth map estimation using the following five evaluation metrics. In the following, $y_{i}$ denotes the depth of the pixel *i* in the estimated depth map, $y_{i}^{*}$ denotes the depth of pixel *i* in the ground-truth depth map, and *T* denotes a set of pixels for evaluation.

The first metric is the scale invariant logarithmic error (SILog) [21], which is defined by
(13)$$\begin{array}{c} \mathrm{SILog} \end{array}:\frac{1}{2∥T∥}\sum_{i\in T}{\left(\log\frac{y_{i}}{y_{i}^{*}}+\frac{1}{∥T∥}\sum_{i\in T}\log\frac{y_{i}^{*}}{y_{i}}\right)}^{2}.$$

This is a metric that evaluates the scale-independent error between the ground truth and estimated depths, where lower values indicate that the estimated depths are correct. For example, in Redwood, the depth map estimated by COLMAP is scale-independent, while the ground truth is millimeter-scale. In our experiments, the scale between the ground truth and the estimated depth map is estimated by the least-squares algorithm and adjusted to the millimeter scale for a fair evaluation. If the estimated depths contain outliers, the scale estimation has errors. SILog evaluates scale-invariant errors and is therefore less sensitive to errors in scale fitting.

The second metric is the Absolute Relative Difference (AbsRel), which is defined by
(14)$$\mathrm{AbsRel}:\frac{1}{∥T∥}\sum_{i\in T}∥y_{i}-y_{i}^{*}∥/y^{*}.$$

This is a metric that evaluates the absolute relative error between the ground truth and the estimated depths, where lower values indicate that the estimated depths are correct.

The third metric is the Squared Relative Difference (SqRel), which is defined by
(15)$$\mathrm{SqRel}:\frac{1}{∥T∥}\sum_{i\in T}∥y_{i}-y_{i}^{*}{∥}^{2}/y_{i}^{*}.$$

This is a metric that evaluates the squared relative error between the ground truth and the estimated depths, where lower values indicate that the estimated depths are correct. SqRel is sensitive to outliers since the larger the error in the estimated value, the larger the evaluated value.

The fourth metric is Root Mean Squared Error (RMSE(log)) on a logarithmic scale, which is defined by
(16)$$\;\mathrm{RMSE}(\log):\sqrt{\frac{1}{∥T∥}\sum_{i\in T}{∥\log y_{i}-\log y_{i}^{*}∥}^{2}}.$$ This is a metric that evaluates the root mean square error between the ground truth and estimated depths, where lower values indicate that the estimated depths are correct.

The fifth metric evaluates the ratio between the ground truth and the estimated depths that is less than the threshold, which is given by
(17)$$\delta <\mathrm{threshold}:\%\;\mathrm{of}\;y_{i}\;s.t.\max_{i}\left(y_{i}/y_{i}^{*},y_{i}^{*}/y_{i}\right)=\delta <\mathrm{threshold}.$$

This indicates that, the larger the value, the more accurate the estimated depth.

The first to fourth metrics evaluate the error between the ground truth and the estimated depths, and the fifth evaluates the accuracy of the estimated depths. As mentioned in the first metric, the depth maps estimated by the conventional and proposed methods are different in scale from the ground truth measured in millimeters. Therefore, except for SILog, the scale of the depth maps has to be aligned when evaluating accuracy. In our experiments, the scale is obtained using the least-squares algorithm so that the error between the sparse depth at each view created from the sparse 3D point cloud estimated by SfM and the corresponding ground truth is small. Using the obtained scale, we evaluate the estimation accuracy by converting the depth maps estimated by each method to the millimeter scale.

### 4.4. Ablation Study of Depth Loss

In this subsection, we describe an ablation study on the depth loss of the proposed method to confirm the dependence of the proposed method on the parameters. In this experiment, we use amp#05668 in Redwood.

#### 4.4.1. Threshold of Huber Loss

The depth loss used in the proposed method is designed based on the Huber loss as described in Section 3.3.2. Huber loss uses L2 loss if the difference between the estimated depth and the true value is less than or equal to the threshold ϵ, otherwise L1 loss is used. Therefore, ϵ has an impact on the accuracy of the depth map estimation. Table 2 shows the accuracy of depth map estimation using the proposed method when Huber loss ϵ is multiplied by the scale factor *s*, where the numbers in bold and underlined indicate the best and second best in each evaluation metric, respectively. In the case of ϵ multiplied by 0.5, i.e., s=0.5, AbsRel, SqRel, and RMSE, the accuracy of the depth estimation is the best, while SILog and δ<1.25 are the third most accurate. In the case of ϵ multiplied by 0.1 and 2, i.e., s=0.1,2.0, the accuracy of the depth estimation is degraded for most of the evaluation metrics. On the other hand, when ϵ is used, i.e., s=1.0, the accuracy of depth estimation is within the top two across all of the evaluation metrics. From the above, the proposed method employs s=1.0 as a scale factor for the threshold ϵ for depth loss.

#### 4.4.2. Hyper Parameter of Objective Function

The objective function used in the proposed method has a hyperparameter $\lambda_{D}$ that adjusts the balance between the color reconstruction loss $L_{Color}$ and the depth loss $L_{Depth}$. In this experiment, we perform an ablation study on $\lambda_{D}$. Table 3 shows the accuracy of the depth map estimation of the proposed method when $\lambda_{D}$ is changed. The accuracy of the depth map estimation of the proposed method is the highest when $\lambda_{D}=0.1$. Therefore, $\lambda_{D}=0.1$ is used in the following experiments.

#### 4.4.3. Difference between Other Depth Loss

In this experiment, we conducted the ablation study for the proposed method using MSE (L2 loss), MAE (L1 loss), and the proposed depth loss based on Huber loss as the depth loss $L_{Depth}$ of the proposed method. We used “amp#05668” from Redwood as input images in this experiment. Table 4 shows the results of the ablation study. As for MSE, the accuracy of depth map estimation is high for AbsRel and RMSE(log), which is comparable to that using the proposed depth loss. As for the proposed depth loss, the accuracy of depth map estimation is high for SILog, SqRel, and δ<1.25. Since the SILog of the proposed depth loss is the highest, the use of the proposed depth loss makes it possible to estimate a smooth and highly accurate depth map. As mentioned above, the evaluation metrics other than SILog are sensitive to the scale between the estimated depth map and the ground truth. The high value of SILog indicates that the estimation accuracy of the depth map is high independent of the scale fitting error. Figure 5 shows the depth maps obtained by each method. In the case of MSE, the object boundary is smooth, although there are some missing areas on the surface of the amplifier. This is because MSE is sensitive to outliers, and the MLP was optimized to be close to the outlier of the depth map estimated by MVS in COLMAP. In the case of using MAE, there is no missing area on the object surface, although the object boundary is not smooth. In the case of the proposed depth loss, there is no missing area on the object’s surface and the object boundary is sharp. As a result, the depth map can be estimated with the highest accuracy using the proposed depth loss.

### 4.5. Comparison with Conventional Methods

This section demonstrates the effectiveness of the proposed method by comparing the accuracy of depth map estimation using the conventional and proposed methods using Redwood and DTU.

Table 5 and Table 6 show the quantitative results for Redwood. COLMAP and NeRF have larger errors and lower accuracy than the other methods, indicating that the depths contain large errors. RC-MVSNet and the proposed method exhibit better results than other methods in most evaluation metrics. In particular, the SILog for the proposed method is smaller than that for COLMAP, NeRF, DS-NeRF, and RC-MVSNet in most cases. This result indicates that the depth map refined by the proposed method contains fewer errors. Figure 6 shows the depth maps estimated by each method. RC-MVSNet shows comparable results to the proposed method in the quantitative evaluation; however, it has more missing regions in the depth map compared to the other methods. The reason for this is that RC-MVSNet uses filtering of the depth map based on reprojection errors. Therefore, the estimated depths are highly accurate, while the depth maps include missing regions. The proposed method estimates the depth map more smoothly than the conventional methods. For example, the proposed method can estimate accurate and smooth depths of flat surfaces such as the floor and the ground in “amp#05668” and “childseat#04134”. This is because the proposed method optimizes the radiance field based on the depth map estimated by COLMAP, unlike NeRF and DS-NeRF. These results indicate that the depth map estimated by COLMAP can be refined through the iterative optimization of an MLP representing the radiance field since the proposed method has fewer missing regions than the depth map estimated by COLMAP. On the other hand, neither COLMAP nor the proposed method could estimate the depth of the surface of the trashcan with poor texture in “trashcontainer#07226”. In “travelingbag#01991”, COLMAP has missing depths for the surface of the traveling bag, while the proposed method smoothly estimated their depths. The difference between “trashcontainer#07226” and “travelingbag#01991” is the size of the missing region in the depth map estimated by COLMAP. If the missing regions in the input depth map are large, the proposed method cannot interpolate the depth map.

Table 7 shows the quantitative results for DTU. The proposed method exhibits better results than the conventional methods in most evaluation metrics. In particular, the SILog for the proposed method is smaller or equal to that for COLMAP, NeRF, DS-NeRF, and RC-MVSNet. The proposed method has few large outliers in the depths since the errors are small and the accuracy is high, as shown in Table 7. Figure 7 shows the depth maps estimated by each method. All of the methods estimated depth maps with high accuracy in DTU. As mentioned in the experimental results for Redwood, RC-MVSNet stands out as having missing regions compared to the other methods. NeRF, DS-NeRF, and the proposed method estimated accurate depth maps even for poor-texture regions compared to COLMAP since the depth maps are synthesized from the radiance field.

In “scan118”, NeRF has small missing regions on the object surface, while DS-NeRF and the proposed method do not have such regions. Since the proposed method has less noise near the object boundaries than DS-NeRF, the complementarity between MVS and NeRF can be utilized to estimate the depth map. On the other hand, the proposed method did not significantly improve the accuracy of depth map estimation for DTU compared to Redwood. This is because the size and number of input images differ between DTU and Redwood. Redwood uses 11 images with 640 × 480 pixels, while DTU uses 49 images with 800 × 600 pixels. The multi-view images in DTU have a sufficient number of viewpoints for depth map estimation and are rich enough in object texture to allow the depth map to be estimated with high accuracy even with conventional methods.

### 4.6. 3D Reconstruction

We reconstructed the 3D point clouds by applying depth map fusion [7,12,31] to the depth maps estimated by COLMAP and the proposed method. In this experiment, we used “Scan9”, “Scan33”, and “Scan118” from the DTU dataset, and multi-view images taken outdoors by the authors.

Figure 8 shows the reconstructed 3D point clouds for COLMAP and the proposed method. Note that the background regions are detected by image segmentation using SAM [32] and are masked in the depth maps to reconstruct the 3D point clouds for better visibility. In “scan 9”, the proposed method has fewer missing regions on the roofs of building, and fewer outliers around chimneys and walls than COLMAP. In “scan33”, COLMAP cannot reconstruct 3D points in the region with poor texture on the headset, while the proposed method can reconstruct 3D points even in such a region. In “scan118”, the proposed method can reconstruct the 3D points in the region where COLMAP cannot. In particular, in “scan118”, the proposed method has a wider reconstruction range than COLMAP. These results indicate that the proposed method can reconstruct regions that cannot be reconstructed by COLMAP by refining the depth map estimated by COLMAP.

We evaluate the applicability of the proposed method by performing 3D reconstruction from multi-view images taken outdoors using an ordinary camera. The dataset consists of 35 RGB images of “Shore to Shore”, which is a 14-foot bronze-cast sculpture located in Vancouver’s Stanley Park, Canada, taken by the authors in June 2023. Figure 9 shows examples of images used in this experiment. It is a difficult situation to apply multi-view stereo and NeRF to since not only the sculpture but also dynamic objects such as tourists are in the image. Figure 10 shows the results of 3D reconstruction from multi-view images using COLMAP and the proposed method. COLMAP reconstructs the details of the sculpture, while there are many outliers on the object’s surface and at the object boundaries. The proposed method reconstructs the sculpture with high accuracy due to there being few outliers on the object’s surface. From the above, the proposed method can refine the depth maps estimated by COLMAP in real-world environments.

## 5. Conclusions

In this paper, we proposed a method to refine the depth maps obtained by MVS through the iterative optimization of an MLP in NeRF. We focused on the fact that MVS can accurately estimate depths in rich-texture regions and NeRF can accurately estimate depths in poor-texture regions and object boundaries, and exploited the complementarity between them. From the viewpoint of NeRF, this approach corresponds to overfitting the MLP with training data, while we conceived of optimizing the MLPs using input images to refine their depth maps. Through a set of experiments using the Redwood-3dscan dataset [19] and the DTU dataset [20], we clearly demonstrated the effectiveness of the proposed method compared to conventional methods. One of the challenging tasks in MVS is to reconstruct the 3D shapes of transparent and translucent objects [33]. The method described in this paper cannot reconstruct the 3D shapes of transparent and translucent objects since the depth map estimated by COLMAP is used. The 3D shapes of transparent and translucent objects can be reconstructed by using photometric stereo, which estimates surface normals from images taken by a camera under varying lighting [34]. NeRF can also consider the degree of transparency on the rays to take into account transparent and translucent objects. We expect that the combination of photometric stereo and the proposed method will be effective in addressing this task. Thus, we will consider refining the depth maps obtained by other MVS using the proposed method and also optimizing the camera parameters by NeRF in our framework.

## Figures and Tables

**Figure 1 jimaging-10-00068-f001:**
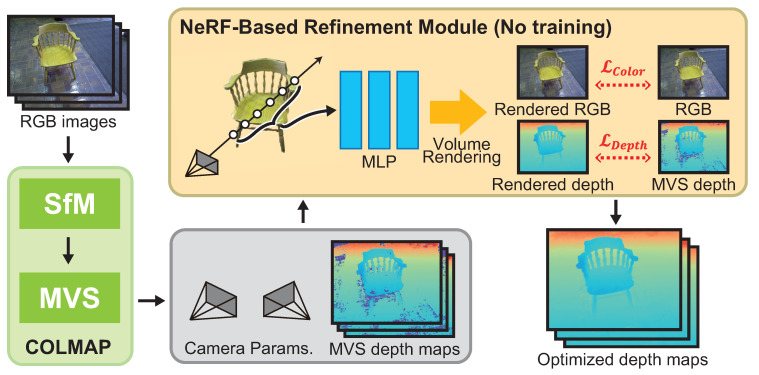
Overview of the proposed method (SfM: Structure from Motion, MVS: Multi-View Stereo).

**Figure 2 jimaging-10-00068-f002:**
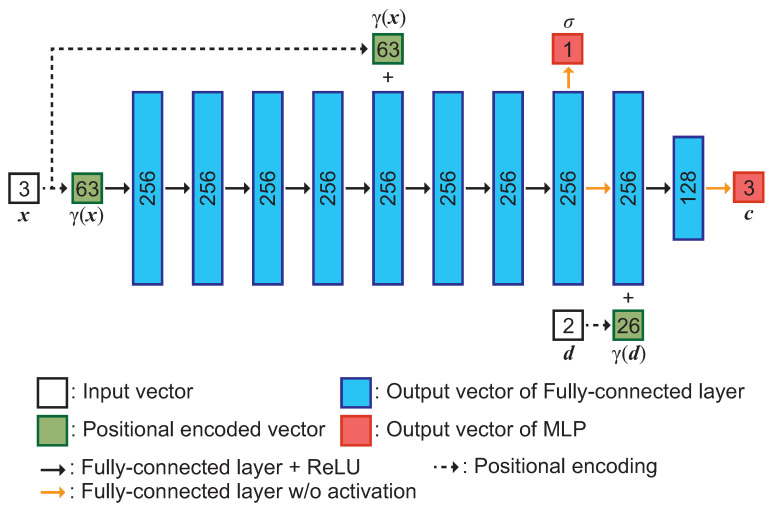
The network architecture of the MLP used in the proposed method, where the number inside the boxes indicates the dimension of each feature vector.

**Figure 3 jimaging-10-00068-f003:**
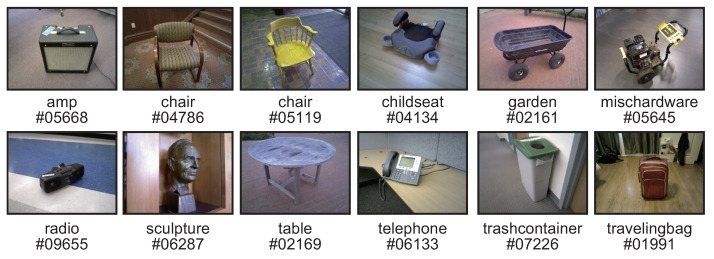
Example of images from Redwood used in the experiments, where images are extracted from the RGB-D video.

**Figure 4 jimaging-10-00068-f004:**
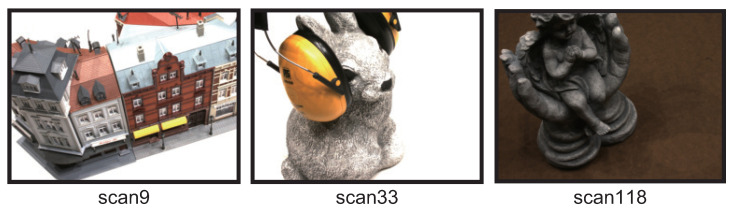
Example of images from DTU used in the experiments.

**Figure 5 jimaging-10-00068-f005:**
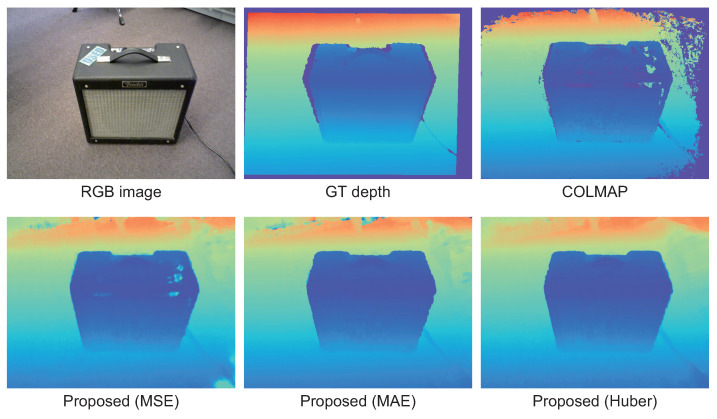
Depth maps estimated by COLMAP and the proposed method with a variety of depth loss functions, where blue in the depth map indicates close to the camera and red indicates far from the camera.

**Figure 6 jimaging-10-00068-f006:**
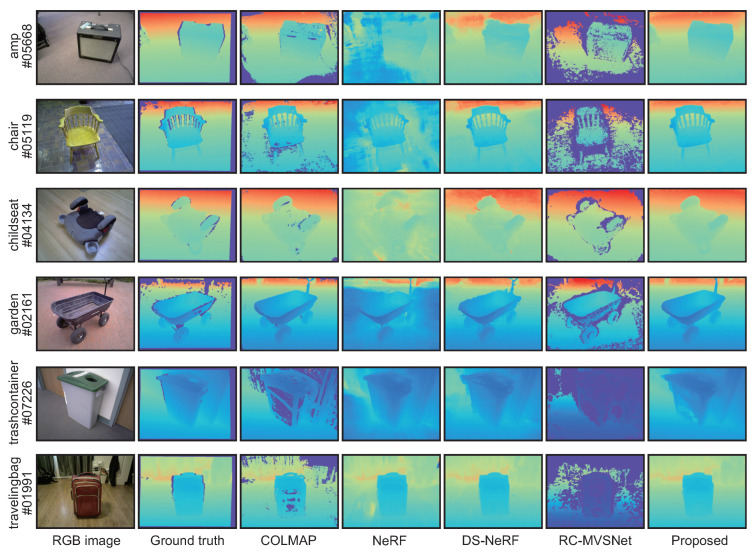
Estimated depth maps for each method in Redwood, where blue in the depth map indicates close to the camera and red indicates far from the camera.

**Figure 7 jimaging-10-00068-f007:**
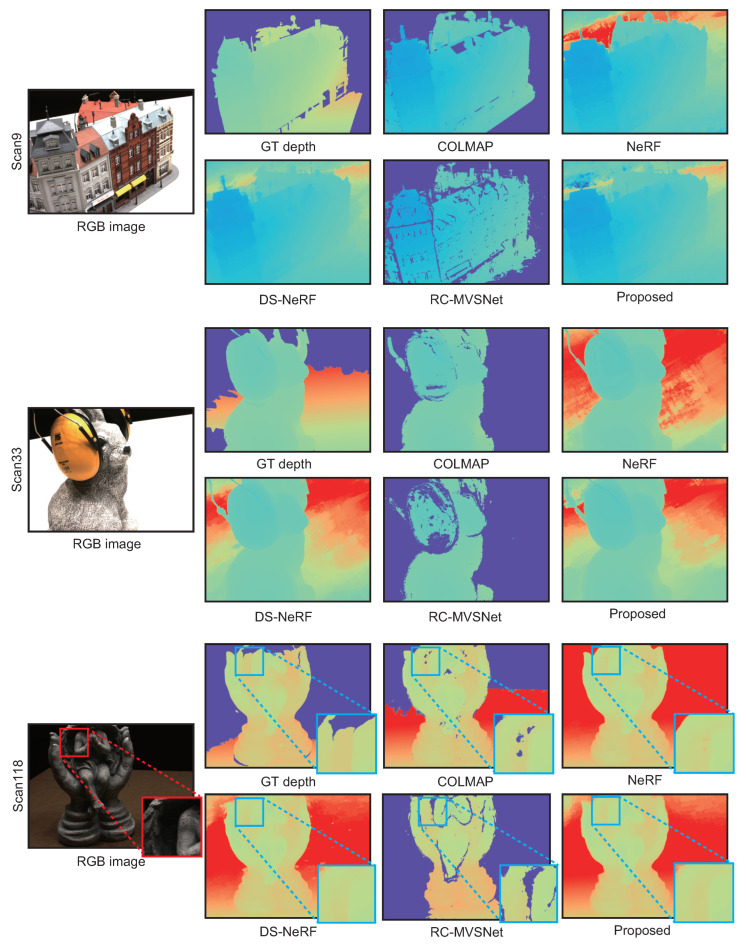
Estimated depth maps for each method in DTU, where blue in the depth map indicates close to the camera and red indicates far from the camera.

**Figure 8 jimaging-10-00068-f008:**
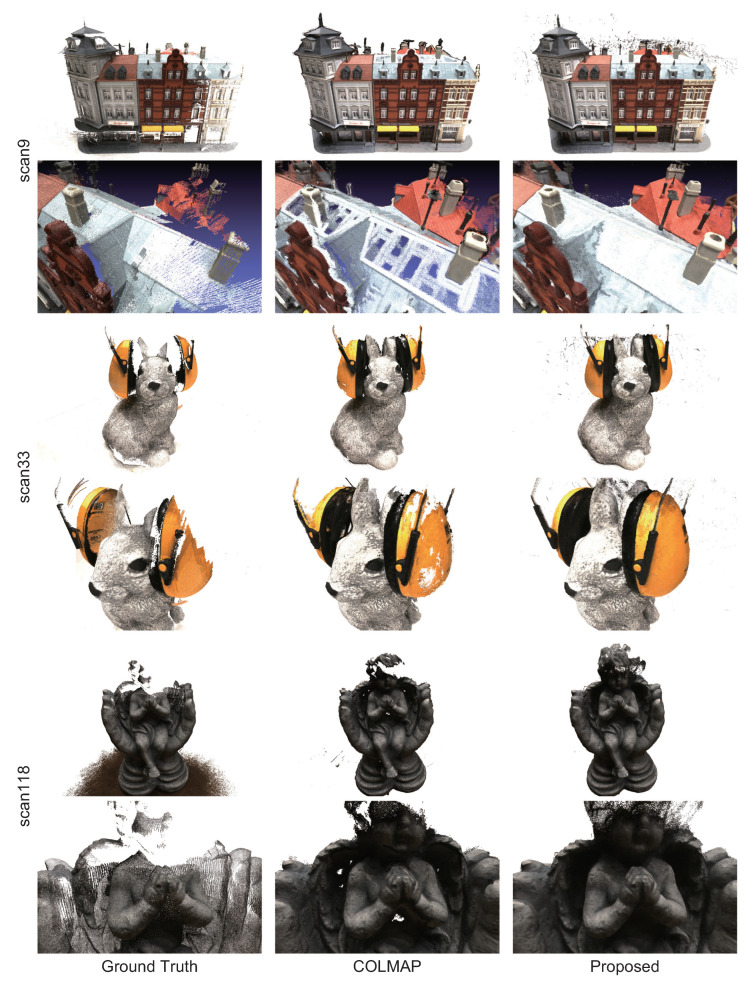
3D point clouds reconstructed from estimated depth maps by COLMAP and the proposed method in DTU.

**Figure 9 jimaging-10-00068-f009:**
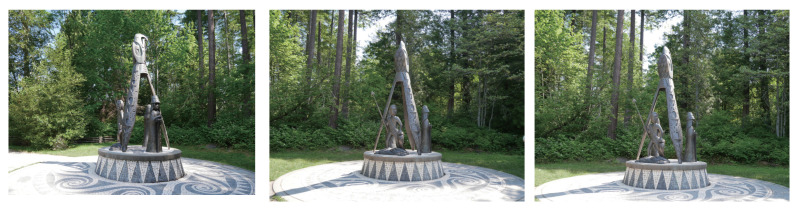
Examples of images of ”Shore to Shore”, which is a 14-foot bronze-cast sculpture located in Vancouver’s Stanley Park, Canada, taken by the authors in June 2023.

**Figure 10 jimaging-10-00068-f010:**
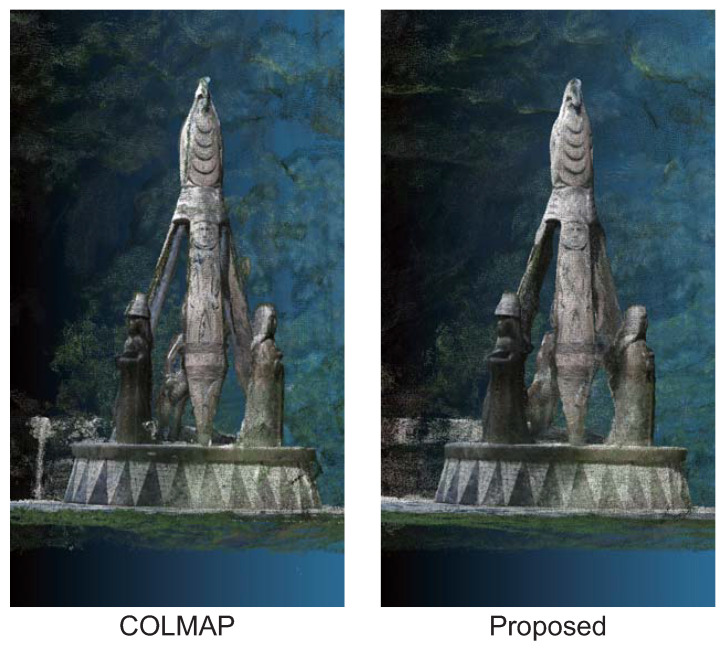
Three-dimensional point clouds reconstructed from depth maps estimated by COLMAP and the proposed method in our dataset.

**Table 1 jimaging-10-00068-t001:** A set of hyper parameters used in the experiments.

Method	# of Iterations		Batch Size	Ratio of Depth Rays	$\lambda_{D}$
	** Redwood [19] **	** DTU [20] **			
	**[Times]**	**[Times]**	**[Rays]**	**[Rays/Batch Size]**	
COLMAP [3]	–	–	–	–	–
NeRF [6]	15,000	100,000	5120	–	–
DS-NeRF [23]	15,000	100,000	5120	0.5	0.1
Proposed	15,000	100,000	5120	0.2	0.1

**Table 2 jimaging-10-00068-t002:** Experimental results of depth map estimation using the proposed method when ϵ of the Huber loss is multiplied by the scale factor *s*. The numbers in bold and underlined indicate the best and the second best in each evaluation metric, respectively. The up arrow indicates that higher values represent better results, while the down arrow indicates that lower values represent better results, respectively.

*s*	Error↓				Accuracy ↑
	**SILog** **[log(mm) × 100]**	**AbsRel** **[%]**	**SqRel** **[%]**	**RMSE** **(log)** **[log(mm)]**	δ<1.25 **[%]**
0.1	0.5626	0.0804	3.135	0.1097	97.61
0.5	0.5787	0.0786	2.990	0.1094	98.51
1.0	0.5759	0.0786	2.992	0.1095	98.53
2.0	0.5970	0.0779	2.988	0.1099	98.55

**Table 3 jimaging-10-00068-t003:** Experimental results of depth map estimation using the proposed method when $\lambda_{D}$ of the objective function is changed. The numbers in bold indicate the best in each evaluation metric.
The up arrow indicates that higher values represent better results, while the down arrow indicates that lower values represent better results, respectively.

$\lambda_{D}$	Error↓				Accuracy ↑
	**SILog** **[log(mm) × 100]**	**AbsRel** **[%]**	**SqRel** **[%]**	**RMSE** **(log)** **[log(mm)]**	δ<1.25 **[%]**
0.05	0.5822	0.0789	3.008	0.1099	98.45
0.1	**0.5759**	**0.0786**	**2.992**	**0.1095**	**98.53**
0.2	0.6086	0.0791	3.032	0.1115	98.44
0.5	0.6180	**0.0786**	3.040	0.1115	98.41

**Table 4 jimaging-10-00068-t004:** Summary of qualitative experimental results in the ablation study for the proposed methods with a variety of depth loss functions. The numbers in bold indicate the best in each evaluation metric. The up arrow indicates that higher values represent better results, while the down arrow indicates that lower values represent better results, respectively.

Depth Loss	Error↓				Accuracy ↑
	**SILog** **[log(mm) × 100]**	**AbsRel** **[%]**	**SqRel** **[%]**	**RMSE** **(log)** **[log(mm)]**	δ<1.25 **[%]**
MSE	0.6413	0.0773	3.025	0.1097	98.05
MAE	0.6623	0.0790	3.107	0.1139	98.24
Huber (Proposed)	0.5765	0.0791	3.015	0.1099	98.47

**Table 5 jimaging-10-00068-t005:** Summary of qualitative experimental results in Redwood. The numbers in bold indicate the best in each evaluation metric. The up arrow indicates that higher values represent better results, while the down arrow indicates that lower values represent better results, respectively.

Datasets	Method	Error↓				Accuracy ↑
		**SILog** **[log(mm) × 100]**	**AbsRel** **[%]**	**SqRel** **[%]**	**RMSE** **(log)** **[log(mm)]**	δ<1.25 **[%]**
amp #05668	COLMAP [3]	6.533	0.0906	4.248	0.2711	97.64
	NeRF [6]	8.313	0.2087	9.784	0.3913	84.64
	DS-NeRF [23]	0.7120	0.0794	3.289	0.1146	**99.40**
	RC-MVSNet [17]	4.637	0.0857	3.175	0.1678	98.64
	Proposed	**0.5759**	**0.0786**	**2.992**	**0.1095**	99.18
chair #04786	COLMAP [3]	12.86	0.1745	8.686	0.3982	95.94
	NeRF [6]	27.22	0.5636	24.51	1.252	40.91
	DS-NeRF [23]	2.895	0.1795	8.638	0.2523	97.22
	RC-MVSNet [17]	**0.9583**	0.1595	**6.554**	**0.1974**	**98.81**
	Proposed	1.315	**0.1556**	6.785	0.1982	98.62
chair #05119	COLMAP [3]	17.45	0.1016	8.400	0.4359	95.28
	NeRF [6]	16.41	0.3130	19.53	0.5966	71.09
	DS-NeRF [23]	1.098	0.0754	5.089	0.1167	**99.51**
	RC-MVSNet [17]	**0.8361**	**0.0656**	**4.396**	0.1054	99.35
	Proposed	1.026	0.0663	4.507	0.1130	99.21
childseat #04134	COLMAP [3]	4.151	0.0539	2.674	0.2009	99.25
	NeRF [6]	3.328	0.1345	5.598	0.1900	99.99
	DS-NeRF [23]	0.1874	0.0527	1.890	0.0624	**100.0**
	RC-MVSNet [17]	**0.1023**	**0.0481**	1.692	**0.0554**	**100.0**
	Proposed	0.1280	0.0488	**1.670**	0.0563	**100.0**
garden #02161	COLMAP [3]	3.916	0.0928	5.647	0.2174	98.47
	NeRF [6]	9.752	0.2282	14.92	0.4288	83.19
	DS-NeRF [23]	0.8502	0.0892	5.165	0.1272	99.33
	RC-MVSNet [17]	**0.4220**	**0.0824**	**4.139**	**0.1032**	**99.73**
	Proposed	0.8336	0.0883	4.969	0.1269	99.09
mischardware #05645	COLMAP [3]	16.25	0.1213	14.77	0.4189	95.09
	NeRF [6]	7.251	0.2172	12.80	0.3726	90.79
	DS-NeRF [23]	1.913	0.1001	5.966	0.1700	99.73
	RC-MVSNet [17]	2.886	**0.0656**	**3.837**	0.1327	99.37
	Proposed	**0.8973**	0.0664	4.030	**0.1137**	**99.74**

**Table 6 jimaging-10-00068-t006:** Summary of qualitative experimental results in Redwood (continued). The numbers in bold indicate the best in each evaluation metric. The up arrow indicates that higher values represent better results, while the down arrow indicates that lower values represent better results, respectively.

Datasets	Method	Error↓				Accuracy ↑
		**SILog** **[log(mm) × 100]**	**AbsRel** **[%]**	**SqRel** **[%]**	**RMSE** **(log)** **[log(mm)]**	δ<1.25 **[%]**
radio #09655	COLMAP [3]	7.308	0.0606	14.98	0.2666	98.07
	NeRF [6]	3.968	0.1430	6.854	0.2345	98.497
	DS-NeRF [23]	1.501	0.0596	4.134	0.1233	99.81
	RC-MVSNet [17]	7.078	0.0350	2.521	0.1702	98.25
	Proposed	**0.2654**	**0.0235**	**1.730**	**0.0520**	**99.99**
sculpture #06287	COLMAP [3]	31.79	0.3789	10.38	0.7903	86.08
	NeRF [6]	6.3333	0.5077	13.431	0.7880	44.12
	DS-NeRF [23]	1.136	0.3292	8.511	0.4179	97.58
	RC-MVSNet [17]	5.573	0.3340	**8.103**	0.4584	96.64
	Proposed	**0.8592**	**0.3230**	8.280	**0.4037**	**98.08**
table #02169	COLMAP [3]	5.456	0.1145	8.168	0.2391	97.53
	NeRF [6]	23.94	0.1973	20.15	0.5058	87.62
	DS-NeRF [23]	5.005	0.1320	10.358	0.2251	95.90
	RC-MVSNet [17]	2.106	**0.1091**	**5.761**	**0.1517**	**99.03**
	Proposed	**1.920**	**0.1091**	6.927	0.1572	98.50
telephone #06133	COLMAP [3]	13.98	0.1245	7.174	0.3890	94.29
	NeRF [6]	16.52	0.2938	13.25	0.5518	75.79
	DS-NeRF [23]	2.935	0.1196	5.848	0.1949	97.87
	RC-MVSNet [17]	7.957	0.0962	**4.651**	0.2602	97.07
	Proposed	**2.412**	**0.0915**	4.909	**0.1676**	**98.19**
trashcontainer #07226	COLMAP [3]	22.14	0.1117	6.877	0.4908	94.15
	NeRF [6]	2.085	0.1083	6.027	0.1874	99.03
	DS-NeRF [23]	0.2313	**0.0563**	2.481	0.0716	**99.99**
	RC-MVSNet [17]	**0.0332**	0.0566	**1.930**	**0.0611**	**99.99**
	Proposed	0.1365	0.0564	2.159	0.0672	99.98
travelingbag #01991	COLMAP [3]	12.18	0.0800	7.401	0.347	95.85
	NeRF [6]	1.401	0.0760	5.071	0.1231	98.86
	DS-NeRF [23]	0.9334	0.0540	3.691	0.090	**99.06**
	RC-MVSNet [17]	29.25	0.1075	7.537	0.4499	92.53
	Proposed	**0.9091**	**0.0487**	**3.497**	**0.087**	**99.06**

**Table 7 jimaging-10-00068-t007:** Summary of qualitative experimental results in DTU. The numbers in bold indicate the best in each evaluation metric. The up arrow indicates that higher values represent better results, while the down arrow indicates that lower values represent better results, respectively.

Scene	Method	Error↓				Accuracy ↑
		**SILog** **[log(mm) × 100]**	**AbsRel** **[%]**	**SqRel** **[%]**	**RMSE** **(log)** **[log(mm)]**	δ<1.25 **[%]**
scan9	COLMAP [3]	6.039	0.3602	9.059	0.5097	98.10
	NeRF [6]	0.8856	**0.3528**	8.793	0.4324	99.72
	DS-NeRF [23]	0.7815	0.3529	8.784	**0.4327**	**99.74**
	RC-MVSNet [17]	13.83	0.3785	9.6952	0.6160	93.40
	Proposed	**0.7280**	0.3530	**8.783**	0.4330	**99.74**
scan33	COLMAP [3]	14.68	0.1064	4.428	0.3971	96.76
	NeRF [6]	1.115	0.0840	3.230	0.1251	99.70
	DS-NeRF [23]	1.034	**0.0837**	3.093	0.1231	99.71
	RC-MVSNet [17]	7.376	0.0935	3.646	0.2886	97.71
	Proposed	**0.9809**	**0.0837**	**3.002**	**0.1219**	**99.72**
scan118	COLMAP [3]	6.932	0.0372	2.968	0.2629	98.61
	NeRF [6]	0.9723	0.0342	3.004	0.0984	99.38
	DS-NeRF [23]	0.7852	0.0302	2.490	0.0888	99.43
	RC-MVSNet [17]	7.091	0.0421	2.996	0.2476	97.69
	Proposed	**0.7282**	**0.0296**	**2.318**	**0.0855**	**99.45**

## Data Availability

Publicly available datasets were analyzed in this study. The Redwood-3dscan dataset can be found here: http://redwood-data.org/3dscan/ (accessed on 7 February 2024). The DTU dataset can be found here: https://roboimagedata.compute.dtu.dk/?page_id=36 (accessed on 7 February 2024).

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
