# Peer review of "Neural Radiance Field-Inspired Depth Map Refinement for Accurate Multi-View Stereo†"

_2313-433X, 2024, doi:10.3390/jimaging10030068_

Round 1
Reviewer 1 Report
Comments and Suggestions for Authors
Dear Editor,
Dear Authors,
The manuscript titled "NeRF-Inspired Depth Map Refinement for Accurate Multi-View Stereo" by Ito et al. presents a method for refining depth maps obtained through Multi-View Stereo (MVS) by iteratively optimizing Neural Radiance Field (NeRF).
The authors propose to combine the strengths of MVS and NeRF in depth map estimation and utilize NeRF specifically for depth map refinement. They introduce a Huber loss into the NeRF optimization process to enhance the accuracy of depth map refinement, particularly in reducing estimation errors in radiance fields. The manuscript is well-written, with clear descriptions of methods and results, making it highly suitable for publication in the targeted journal. It was a real pleasure to read it!
However, there are certain considerations that warrant attention.
(1) Firstly, while the manuscript addresses depth map refinement effectively, there may be limitations in its applicability to datasets with high degrees of transparency or translucency. A discussion on how the proposed method handles such cases could enhance the manuscript's relevance and broaden its audience, considering transparency is a significant challenge in Multi-View Stereo.
(2) Secondly, despite the acknowledgement of the manuscript being an extended version of a previous publication, there is a notable similarity percentage (23%) that suggests room for further revision to minimize overlap. It is recommended that the authors carefully review the manuscript to ensure distinctiveness from the previously published work and make necessary adjustments to reduce similarity.
In summary, the manuscript presents an innovative approach to depth map refinement in Multi-View Stereo, offering valuable contributions to the field. Addressing the issues related to transparency considerations and reducing similarity with the previous publication would enhance the manuscript's impact and appeal to a wider audience.
Best regards.
Author Response
Thank you for your careful review of our papaer.
We attached the PDF file to answer your comments.

Reviewer 2 Report
Comments and Suggestions for Authors
This paper introduces a novel technique that refines depth maps from Multi-View Stereo (MVS) via iterative optimization of Neural Radiance Fields (NeRF). It synergizes the object surface depth estimation capabilities of MVS with the boundary depth estimation strengths of NeRF, incorporating Huber loss to enhance the precision of depth map refinement. Experiments conducted on the Redwood-3dscan and DTU datasets demonstrate the method's superiority over traditional approaches such as COLMAP, NeRF, and DS-NeRF. Moreover, it utilizes depth scale-invariant metrics alongside standard depth map estimation measures to validate its effectiveness. However, the study has certain shortcomings and areas that require clarification:
- Experimental Design and Dataset Diversity: The research primarily utilizes Redwood-3dscan and DTU datasets for validation. To enhance the universality of the findings, it is recommended that the authors expand the experimental scope to include a wider variety of datasets, assessing the method's adaptability and robustness across different scenarios.
- Comparative Analysis: While comparisons have been made with traditional methods like COLMAP, NeRF, and DS-NeRF, there is a lack of direct comparison with the latest related works. The authors are advised to update the literature review and compare their method against the state-of-the-art to highlight its uniqueness and advantages.
- Methodological Details and Transparency: To improve reproducibility, more detailed descriptions of the method, including parameter settings, network architecture, and optimization strategies, are suggested. Moreover, open-sourcing the code and datasets, where possible, would significantly enhance the impact of the work.
- Limitations in Depth Map Refinement: The research focuses on depth map refinement but inadequately addresses challenges such as occlusions and complex textures. It is suggested that the authors delve deeper into these challenges and propose potential solutions or directions for improvement.
- Parameter Sensitivity Analysis: Including a series of charts that show how main parameters (e.g., iteration numbers, Huber loss thresholds) affect model performance would help understand the model's sensitivity to parameters and guide the setting of parameters in practical applications.
- Case Studies and Visual Results: Adding visual result comparisons for specific case studies, especially in complex scenarios (e.g., varying lighting conditions, occlusions), would visually demonstrate the method's advantages and limitations.
- Practical Applications and Case Studies: Although the method performs well on selected datasets, there is a lack of case analysis in real-world application scenarios. Providing examples of specific applications, such as cultural heritage reconstruction or virtual reality, would help readers better understand the method's potential value and application range.
These suggestions aim to further enhance the study's academic value and practicality, providing valuable reference for future research.
Comments on the Quality of English LanguageThe English quality is satisfactory, adhering to the standard conventions of academic writing. However, minor improvements could be made in terms of proofreading to correct occasional grammatical errors and enhance clarity in certain sections.
Author Response
Thank you very much for your careful review of our papaer.
We attached the PDF file to answer your comments.

Reviewer 3 Report
Comments and Suggestions for Authors
This paper deals with the problem of estimating the depth maps of a given image. The novelty is the complementary combination of Multi-View Stereo and Neural Radiation Field. Its efficiency is proven by comparing with the conventional methods against public datasets. The paper itself is written well with its strengths and weaknesses. The reviewer can find values with this paper.
The reviewer recommends the following for improving the paper, especially for the readers in other research fields.
1. line 22:
Please define or refer to some papers on the definition of "similarity of textures." How is “similarity” measured?
2. line 23:
In the paper, the word "texture" is used several times, and moreover, "rich texture", "poor-texture" and "low-texture" are used. Please add some explanations or definitions on "texture" for the readers in other research fields. They may be confused with such expressions as "poor," "low" and "object boundary."
3. line 30: On "PatchMatch-based methods assign depth."
How is the depth assigned to each pixel? The reviewer is wondering if the depth is an integer or a floating number. Please add some descriptions on the implementation of depth assignment of PathMatch-based methods.
4. line 48:
Please add a brief explanation on "radiance field."
5. line 82:
The reviewer doesn't understand what the structure means. Moreover, the word "motion" is not clear. Of course, the corresponding paper is referred to, but SfM is used many times in the latter paper, and it is mentioned that SfM is a de facto standard method for estimating the camera parameters. Please add some descriptions on Structure from Motion (SfM).
6. line 93:
The word "NCC with bilateral weights" is used, however, "NCC" is not defined. Please add the description or definition on NCC.
7. with Figure 2:
The explanation is needed for x=3, gamma(x)=63, d=2, gamma(26). sigma=1 and c=3. The reviewer doesn't understand what these values mean, these are examples? and the calculation of gamma( ).
8. line 286:
Typographical error? "In out experiments," --> "In our experiments,"
9. The difference between training and optimizing of MLP
"Optimization of MLP" or similar expressions are used several times. However, the reviewer doesn't understand "optimization of MLP." Usually, MLP is trained, however, the proposed method adopts such MLP that is not trained but optimized. Please describe the optimization of MLP in detail.
Author Response
Thank you very much for providing important comments
We attached the PDF file to answer your comments.
